# Three-Year Follow-Up of the POIBA Intervention on Childhood Obesity: A Quasi-Experimental Study

**DOI:** 10.3390/nu13020453

**Published:** 2021-01-29

**Authors:** Francesca Sánchez-Martínez, Silvia Brugueras, Gemma Serral, Sara Valmayor, Olga Juárez, María José López, Carles Ariza

**Affiliations:** 1Agència de Salut Pública de Barcelona (Public Health Agency of Barcelona, ASPB), Pl Lesseps 1, 08023 Barcelona, Spain; sbruguer@aspb.cat (S.B.); gserral@aspb.cat (G.S.); svalmayo@aspb.cat (S.V.); ojuarez@aspb.cat (O.J.); mjlopez@aspb.cat (M.J.L.); cariza@aspb.cat (C.A.); 2Institut d’Investigació Biomèdica Sant Pau (IIIB Sant Pau), Sant Antoni Maria Claret 167, 08025 Barcelona, Spain; 3Pompeu Fabra University (UPF), Health and Experimental Science Department, Doctor Aiguader 80, 08003 Barcelona, Spain; 4Ciber de Epidemiología y Salud Pública (CIBERESP), Monforte de Lemos 3-5, 28029 Madrid, Spain

**Keywords:** childhood obesity, school-based program, effectiveness evaluation, adiposity, healthy eating, physical activity

## Abstract

Childhood obesity has increased worldwide over the past four decades. This quasi-experimental study aimed to assess the effectiveness of a multicomponent and multilevel school-based intervention (POIBA) at 3 years of follow-up. The nutrition intervention focused on food groups, food pyramid, nutrients, portions, and balanced menus. In total, 3624 children participated in the study. Anthropometric measurements and information on food frequency and behavior, physical activity, and daily screen use were collected in the intervention (IG) and comparison group (CG). Positive unadjusted changes toward adherence to recommendations were found for water, meat, sweets, and fried potato consumption, proper breakfast, not having dinner in front of the TV, out-of-school physical activity, and daily screen use. Three scores were used to calculate the proportion of children making more than one change to improve healthy habits regarding physical activity (global Activity score), nutrition (global Nutrition score), and both (global score). Students exposed to the intervention had a significantly better global Activity score (16.2% IG vs. 11.9% CG; *p* = 0.012) and Global score (63.9% IG vs. 58.5% CG; *p* = 0.025). Intervention effects on obesity incidence at 3-year follow-up lost significance but maintained the positive trend. In conclusion, school-based interventions including a family component could be useful to address the childhood obesity problem.

## 1. Introduction

Childhood obesity has become a major public health problem. It is an independent risk factor for adult obesity and is related to physical, social, and psychological consequences [1]. According to the World Health Organization (WHO), the prevalence of overweight and obesity among children and adolescents aged 5–19 has risen dramatically from 4% in 1975 to over 18% in 2016, with little differences between girls (18%) and boys (19%). Regarding obesity, it has increased from under 1% of children and adolescents aged 5–19 years old in 1975 to 6% in girls and 8% in boys in 2016 [2]. Over the past four decades, the mean body mass index (BMI) and obesity in the child and youth population aged 5–19 years old have increased in most regions and countries worldwide. This rise in children’s and adolescents’ BMI has plateaued in many high-income countries but BMI remains at high levels. In the 2016 ranking of the Organization for Economic Co-operation and Development countries, United States (21.4%), Saudi Arabia (17.4%), and Argentina (16.9%) had the highest prevalence of obesity in children aged 5–19 years old. On the contrary, India (2.0%), Japan (3.3%), and Switzerland (5.8%) presented the lowest. In this ranking, Spain was placed in the 20th position for children aged 5–19 years old with obesity, with a prevalence of 10.8% overall, and 13.1% for boys and 8.4% for girls. Within European Union countries, Spain occupied the 7th and the 4th position, respectively [3].

Overweight and obesity are caused by multiple complex determinants. Social factors are clearly associated with these problems, with some of the most closely related factors being socioeconomic status (SES), country of origin and type of family [4]. Genetic and family factors also have a clear link, with important factors being maternal overweight during pregnancy and not being breastfed [5]. Likewise, lifestyle-related factors such as eating behavior, the practice of physical activity, screen time, and sleep habits play a key role [6,7]. 

The flattening of BMI values in children and adolescents in high-income countries could be explained by specific initiatives carried out by governments, community groups and schools that have increased public awareness about overweight and obesity in children, leading to changes in nutrition and physical activity that can curb the rise in mean BMI [8]. Schools have been proven to be a successful setting for interventions aimed at improving healthy habits, since children spend a large part of their time there [9]. The most effective school-based interventions are multicomponent initiatives, based on the social cognitive model [10] and with environmental implications [11], especially including the family environment [12,13]. 

The two most widely studied behavioral determinants in school-based preventive interventions are food and nutrition and physical activity. Most studies have reported the effectiveness of physical activity interventions [14,15,16], especially those with a longer duration [17] and those with a community component [18]. Other studies, while highlighting the effects on physical activity, focus on the complementary effects of sedentary habits and screen use [12,19]. The Project Healthy Schools is an example of a combined educational strategy that aimed to increase intake of fruits and vegetables, to decrease intake of high-sugar beverages and fatty food, to perform at least 150 min of exercise per day, and to decrease screen time. It showed an increase in the consumption of fruits and vegetables and in the participation in moderate and vigorous exercise activities, and a decrease in time spent in front of screens [19]. 

Most studies use BMI to determine obesity in children and adolescents [14,20,21,22]. However, there is difficulty to establish a clear criterion for young population when defining overweight and obesity through the BMI, since it cannot distinguish between lean and body fat. For that reason, some authors assure that distribution of body fat is a more sensitive indicator to determine body fatness in children and adolescents than BMI. The percentage of body fat can be calculated based on height, weight, and thickness measurements of four skinfolds (biceps, triceps, subscapular, and suprailiac) [23]. In case only one skinfold can be measure, the triceps skinfold is recommended [24].

Follow-up studies in school interventions are scarce, as is an analysis by subgroups such as gender. [25]. As far as we know, among the few existing studies, there are two 8-year follow-ups, showing that the effect on BMI was maintained [22,26], one of them only in children from advantaged socioeconomic backgrounds [26]. However, none of them described the effect of the intervention on physical activity and nutritional habits or the relationship between the type of implementation and the effect of the intervention. The aim of this study was to assess the effectiveness of a childhood obesity prevention intervention at 3 years of follow-up.

## 2. Materials and Methods 

### 2.1. Study Design 

This prospective quasi-experimental study with a comparison group (CG) assessed the effectiveness of a preventive intervention at 3 years, the childhood obesity prevention intervention in Barcelona (POIBA, Prevención de la Obesidad Infantil en Barcelona), 

### 2.2. Study Population

Participants consisted of children born in 2002 who attended school in Barcelona and were aged 8 to 9 years at baseline. A total of 104 schools were selected to participate in the study, of which one in the intervention group (IG) and six in the CG declined to participate. Finally, the IG consisted of 46 schools (74 classrooms), including 1633 children. In all, 1991 children attending 51 schools (88 classrooms) participated in the CG. The schools in the IG were randomly selected in four of the 10 districts of the city, while schools in the CG were selected from the remaining six. The selection was based on established criteria of representativeness according to the type of school and the SES of the neighborhood, with oversampling of disadvantaged neighborhoods. Further details of the study design, study population, setting, and procedure have been previously described in the study protocol of the intervention [27].

#### Intervention

POIBA is a school-based intervention with family involvement. Previous results of the evaluation of the intervention at 1-year follow-up have already been published [28].

Students in the IG took part in the POIBA intervention. This program is a multicomponent (classroom, physical activity and family) and multilevel (individual, school and family) initiative. It consisted of a core intervention, carried out in the fourth grade of primary school (2011–2012 academic year). Some schools also performed a booster session 2 years later, in the sixth grade (2013–2014 academic year). The intervention has previously been described in detail in the published study protocol [27]. 

The three components of the intervention are described below:Classroom component

It was implemented by classroom teachers along the academic year, previously trained by the Public Health Agency of Barcelona. Throughout the year, the teachers had the support of the staff from the research team to solve doubts and problems. 

The intervention “We grow up healthy” consisted of 9 sessions, including 58 activities, classified into 3 modules [29]. Module 1 worked on growth (weight and height) and body image assessment through 2 sessions (15 activities). Module 2 focused on food and nutrition: digestion, food groups, nutrients and their functions, the food pyramid, and proper breakfast through 5 sessions (31 activities). Module 3 worked on physical activity and rest and synthesis of the intervention through 2 sessions (12 activities). 

Module 2, focused on food and nutrition, was designed by professionals with extensive experience in designing school programs to promote healthy lifestyles, especially in the field of nutrition. It consisted of 5 sessions:”Our digestion”. The contents of the session were based on knowing the phases of the digestion process, assessing the importance of chewing food properly, and respecting the necessary rest after each meal to promote the digestion process. The activity worked included a worksheet to complete and a group discussion.“Food groups”. The aims of this session were to know the five food groups (fruit and vegetables; starchy food; dairy; protein (meat, fish, eggs and nuts); oils, fats, and sweets) and to identify the origin of each food (animal, vegetable and mineral). Two activities were proposed: the completion of a worksheet to delve into the different food groups and bringing from home products and packaging of the different food groups.“Nutrients and their functions”. The objectives were to know what nutrients are, identify where nutrients are found, learn to classify foods according to the main functions of nutrients, and learn some dietary recommendations. Two activities were carried out: discussing the contents of a worksheet in the group class and working on some videos focused on nutrients. The main contents of the activities focused on nutrients as essential substances for life; the classification of nutrients, according to their main function, into builders, energy and regulators, and the importance of a varied and balanced diet to obtain the nutrients necessaries for life.“Food pyramid”. The main aims were to know the concept of portion and how the portions are distributed, identify the food groups that form the food pyramid and their frequency of consumption, develop favorable attitudes and basic skills to incorporate the recommended foods into the diet and reduce those of sporadic consumption. The activity consisted of building a big pyramid in the classroom using the products and packaging provided by children and displaying it in a visible place in the classroom.“The best breakfast”. The aims were to identify the three food groups that should be included in a healthy breakfast (dairy, carbohydrates and fruit), realize that breakfast at home and at school are complementary and practice how to make a healthy and desirable breakfast. The activity consisted of preparing a healthy breakfast in small groups with food that children brought from home.

The reinforcement “We grow up more healthy” complemented the work done two years before on food and nutrition during the intervention through 2 sessions, including 8 activities [30]. These sessions consisted of:“Remember the food pyramid”. The main objectives were to review the basic contents of the food groups and their situation in the food pyramid, understand the concept of portion, promote a favorable attitude towards a balanced diet and recognize the nutritional needs of our body. The activity consisted of rebuilding a food pyramid and working on the concept of portion based on the food groups in the pyramid.“A balanced diet”. The objectives were to identify correct and improvable aspects of a weekly menu, develop skills to identify if the frequencies of food groups in a diet are correct, and make proposals for a balanced diet at family dinners. The activity consisted of the assessment in small groups of a balanced weekly menu, making sure that each food group appears in the appropriate quantities and frequencies.

Physical activity component

It was implemented by physical education teachers also with the support of the research team. It consisted of promoting the practice of physical activity among children both at school and after school or in free time. Extracurricular activities were funded in families with financial difficulties. In the reinforcement, two years later, a session was carried out, which consisted of recording the physical activity carried out by each child throughout a month, collecting information from both weekdays and weekends.

Family component

The participation of families in the intervention was carried out at various levels, since they were offered to participate in activities at home together with their children, in free sporting events on weekends and in the workshop “A plan for change” together with their children. This workshop aimed to provide families with skills and strategies to achieve a healthy and balanced diet, improve their sleep hygiene and use of leisure time and reduce screen time. The nutritional content consisted of a review of family habits regarding breakfast and dinner, a discussion among all families attending the workshop about what improvements they should make, and finally, an agreement from each family on a commitment to improve a specific family habit within the next four weeks. 

The reinforcement was focused on the workshop “Let’s improve family health” with the aim of negotiating and agreeing on improvements in meals and physical activity at the weekend. The activity on nutrition consisted of each family agreeing on a balanced dinner that would complement the weekly menu children had previously worked on in the classroom. Likewise, each family should reach an agreement related to healthy eating. 

### 2.3. Data Collection

Measurements were taken in three time periods in both groups:Baseline measurement (T1): Between April and June 2011, in the third year of primary school (8–9 years old).One-year follow-up measurement (T2): Between April and June 2012, in the fourth grade of primary school (9–10 years old).Three-year follow-up measurement (T3): Between April and June 2014, in the sixth grade of primary school (11–12 years old).

A confidential *α*-numeric code was created for each schoolchild, which allowed linking of the questionnaires and the anthropometric measurements carried out in the three time periods.

#### 2.3.1. Anthropometric Measurements

Weight, height, triceps skinfold thickness, and waist and hip circumference were measured at school by trained professionals, with prior informed consent of their families. From these data, the triceps skinfold thickness and BMI were calculated, which allowed information on obesity to be obtained.

#### 2.3.2. Information on Behaviors, Knowledge, and Attitudes

Data were collected through two validated self-administered computerized questionnaires, “POIBA-How do we eat?” for food frequency and food behavior and “POIBA-How do we move?” for physical activity and screen use. Information on sociodemographic data was included in both questionnaires. Students completed the questionnaires in the classroom, under the guidance and supervision of their teachers, who had previously been trained by staff from the Public Health Agency of Barcelona (ASPB). 

Since the questionnaires had not been previously validated, we compare the results of the “POIBA-How do we eat?” questionnaire with a 24 h recall (YANA-C computer program) as the gold standard and a questionnaire addressed to children’s families [31]. In the case of the questionnaire “POIBA-How do we move?”, a family’s questionnaire and a 7-day recall of physical activity were used as indirect methods and accelerometers as a direct method (gold standard) [32].

### 2.4. Variables

#### 2.4.1. Outcome Variables

##### Primary Outcome

The primary outcome was the cumulative incidence rate (CIR), defined as new cases of obesity between baseline and the 3-year follow-up. This variable was constructed from the triceps skinfold values and was calculated according to percentiles of the National Health and Nutrition Examination Survey (NHANES) of the Center for Disease Control and Prevention (CDC) [33]. Based on these percentiles, obesity was defined as thickness values ≥90th percentile for age and sex. The BMI calculation was based on kg/m^2^. For each age and sex, overweight was defined as ≥1 SD from the mean BMI value and obesity as ≥2 SD according to the World Health Organization (WHO) *z* score [34,35].

##### Intermediate Outcome

Intermediate outcome variables were also calculated based on the information obtained from questionnaires. These were:Food frequency: Healthy foods (water, fruits and vegetables, and meat) were measured according to the number of weekly or daily servings. Unhealthy foods (sweets and fried potatoes) were measured according to higher occasional or casual consumption [36].Food behavior: Having a proper breakfast, including carbohydrates, dairy, and fruits [36] (yes/no); having dinner in front of the TV (yes/no); eating at school (yes/no); not eating alone, meaning having breakfast, lunch or dinner with someone at home (yes/no); frequenting fast-food restaurants (yes/no).Physical activity: Out-of-school physical activity (yes/no); engaging in leisure-time physical activity (<2 days/week, ≥2 days/week) [37].Daily use of screens: On weekdays and on weekends (<2 h/day, ≥2 h/day) [38].

Three scores were defined to calculate the percentage of children showing more than one positive change in the indicated variables:Global Nutrition score: More than one change in the nine variables of food frequency and food behavior.Global Activity score: More than one change in the four variables of physical activity and daily screen use.Global score: More than one change in the thirteen variables of the global nutrition score and global activity score.

#### 2.4.2. Independent Variables

Sociodemographic information was collected in both questionnaires. The individual variables collected were age; sex; family structure (living with both parents/single parent/other situations); country of origin (natives when parents and children were born in Spain/immigrants when parents or both generations were born outside Spain), and SES measured through the Family Affluence Scale (FAS). The FAS is an indicator that categorizes the student family’s socioeconomic position as low, middle, and high depending on their answer to four items (car ownership, having their own unshared room, the number of computers at home, and how many times they spent on holidays in the past 12 months) [39]. Contextual variables were type of school (public/private-subsidized) and gross disposable household income (GDHI) (disadvantaged SES for <85/advantaged SES for ≥85) [40]. 

#### 2.4.3. Process Evaluation Variables

Students in the IG were classified according to how the intervention had been implemented in the classroom, the physical education time, and the families’ workshop. The classroom was taken as the sample unit. 

The main variables studied were fidelity to the protocol, number of completed activities in the classroom, children’s physical activity level, and attendance at the family workshop. Based on these three components, a global indicator for the intervention was constructed, with a proportional weight according to the importance of each component of the program. The indicator allowed classification of the intervention into three categories: not acceptable, acceptable, and qualified. A detailed description of the process variables can be found in the previously published article [28].

### 2.5. Ethics

The project was evaluated and approved by the Clinical Research Ethics Committee of Parc de Salut Mar (CEIC-Parc Salut Mar), reference number 2009/3470/l. During the study, national and international guidelines were followed (codes of professional ethics, the Helsinki Declaration of 1964 and subsequent revisions). The Spanish law on data confidentiality was observed (Law 15/1999 of 13 December on Personal Data Protection). Families provided signed consent for their children to participate in the project. All the information was protected by using standard data management and information storage procedures.

### 2.6. Statistical Analysis

A descriptive and bivariate analysis was performed to study the differences between the IG and CG, including complete and incomplete cases. The intermediate outcomes and the sociodemographic and contextual variables were compared in the three time periods. Percentages (for qualitative variables) were compared using the chi-square test and means (for quantitative variables) were compared by using the Student *t*-test. Unadjusted (difference in percentages) and adjusted (odds ratios (OR) and 95% confidence intervals (95% CI)) changes in the intermediate outcomes were studied. The CIR of obesity was calculated by the degree of intervention (CG, IG (total, acceptable and qualified intervention)). This association was obtained using logistic regression models with their OR and 95% CI. The CIR-related variables of obesity were included in a multivariate analysis with adjustment for possible confounding variables. Cohen’s criterion was used to estimate the size effect of the differences. Statistically significant differences were considered at *p*-value <0.05.

Generalized estimating equations analysis was used to adjust for intergroup and intragroup variability and to control the correlations among responses. In the model, the classroom was established as a random effect.

The analysis was performed using the Stata statistical package, version 15.

## 3. Results

The flowcharts of children throughout the 3 years of the study are shown in Figure 1 for the IG and in Figure 2 for the CG. The initial total number of eligible participants was 1633 in the IG and 1991 in the CG. At baseline, anthropometric measurements and questionnaire responses were obtained from 1464 pupils in the IG and 1609 in the CG. After 1 year, follow-up was achieved in 1184 (80.9%) pupils in the IG and 1308 (81.3%) in the CG. At 3 years, follow-up was achieved in 1151 (78.6%) pupils in the IG and 1230 (76.5%) in the CG. The main causes of individual losses to follow-up were; being absent on the day of data collection, being transferred to another school, or having incomplete data, such as birthdate or the confidential code assigned to the student. In addition, 2 schools in the IG and 1 in the CG decided to stop participating in the project, which meant the loss of follow-up of all their children. Finally, of those students who could be followed up, 772 (52.7%) in the IG and 881 (54.8%) in the CG had complete matched data available over the 3-year follow-up, that is, anthropometric measurement and questionnaires responses from the three data collection periods (T1, T2, and T3).

Table 1 shows the main sociodemographic characteristics at baseline of the 2681 children who could be followed at 3 years in the two groups, 1151 in the IG and 1230 in the CG. At baseline, the total 3-year follow-up sample of the IG and CG differed in the GDHI index. Among students lost to follow-up, losses among students attending public and advantaged SES schools were higher in the IG than in the CG.

Primary and intermediate outcomes at the beginning of the study, as well as losses to follow-up, are shown in Table 2. At baseline, 14.8% of the IG and 12.4% of the CG were obese, based on BMI, with no significant differences. Students in the IG less often frequented fast-food restaurants (84.9 vs. 87.0%), practiced less out-of-school physical activity (69.9 vs. 76.5%) and showed more screen time on weekends (58.4 vs. 53.7%) than students in the CG, with these differences being statistically significant.

Table 3 shows a comparison of the data obtained through the baseline questionnaires (T1) with those of the first follow-up after 1 year (T2) and those of the second follow-up at 3 years (T3). The sample in this table included those children who were followed up for 3 years. Regarding the comparison between the information collected at baseline (T1) and after 3 years of follow-up (T3), positive unadjusted changes are shown in drinking water during meals and consumption of meat, occasional sweets, and occasionally fried potatoes in the IG, compared with the CG. For food behavior, there were also positive unadjusted changes in having a proper breakfast and having dinner in front of the TV. For physical activity, greater positive unadjusted changes were found in practicing out-of-school physical activity and daily screen use on weekdays and on weekends. In all the variables analyzed, the differences between adjusted groups were nonsignificant. In terms of scores, significant differences were found between the two groups in the global activity score (16.2% IG vs. 11.9% CG; *p* = 0.012) and in the global score (63.9% IG vs. 58.5% CG; *p* = 0.025), with children in the IG being those with higher scores. No differences were found in the global nutrition score (44.3% IG vs. 41.1% CG; *p* > 0.05).

The incidence of obesity based on triceps skinfold thickness (adiposity) at 3 years of follow-up is shown in Table 4. Data are presented by groups and sex and by the level of implementation of the intervention. Overall, the global incidence of obesity after 3 years was 4.3%, with a significantly higher incidence of obesity in girls than in boys (5.5 vs. 3.1). Within the IG, the incidence was also higher in girls than in boys (6.2 vs. 3.1) but this difference was not statistically significant. Depending on the level of implementation of the intervention, a gradient in the incidence of obesity was observed, both globally in boys and girls, with a higher incidence of obesity in children who did not receive an acceptable intervention and a lower incidence in those who received a qualified intervention. Children from the IG who received a qualified intervention showed an incidence difference of 2.4, which was not statistically significant compared to the CG. Qualified intervention led to a 60.5% reduction in the incidence of obesity, avoiding almost two out of three cases. The Cohen’s *d* effect size of the intervention was 0.67. No differences were found for BMI in the effect of the intervention between the IG and the CG (data not shown).

## 4. Discussion

The evaluation of the POIBA intervention at 3 years of follow-up showed that the proportion of children who made more than one positive change to increase their healthy habits was significantly higher among those who underwent the POIBA intervention, with significant differences in the global score. Furthermore, the global activity score showed significant differences in the IG compared to the CG. Nevertheless, for the global nutrition score, the significant differences found at 1-year follow-up were not maintained at the three-year follow-up. Regarding the incidence of obesity, a positive but nonsignificant trend was found for triceps skinfold in the effect of the intervention when it was implemented in a qualified way.

According to our results, participating in a school-based intervention could be useful to promote improvements in healthy habits among the child population. Actions encouraging physical activity and a reduction of sedentary habits seem to be the most effective initiatives in the school environment, rather than nutritional training. However, the literature on this topic is inconclusive, as there are discrepancies on which components are the most effective. Some authors have reported results in line with our own, highlighting that interventions containing physical activity as a core component or as a single-component show the greatest benefits [16,17]. Nevertheless, other studies have emphasized that interventions with a nutritional component [41] or with combined diet and physical activity components had the greatest effectiveness [14,22,42]

Children in classrooms where the implementation of the intervention was qualified according to the established protocol showed a 60.5% reduction in the incidence of obesity, meaning that almost two in three cases could be avoided. Unlike the results at the 1-year follow-up, when the effect of the intervention was significant [28], the effect diminished at the 3-year follow-up, losing significance but maintaining the positive trend. As in our study, the loss of effect of interventions over time has been previously reported in the literature. A review of school-based interventions concluded that no persistence of positive results in reducing obesity in school-age children was observed [43]. In the same way, a school-based intervention aimed at reducing the prevalence of obesity and increasing healthy lifestyles showed that early effects observed from the longitudinal 2-year were not maintained over 8 years [44]. On the other hand, some studies have reported favorable and sustained effects at 8-year follow-up, such as the Avall Study [22] and the Kiel Obesity Prevention Study [26], so that, more long-term follow-up studies of short-term interventions are needed to determine if the effect of the intervention is maintained over time. 

BMI is a clear indicator of weight problems in adults, but in the case of children there is no clear criterion to define overweight and obesity. The percentage of body fat may be a more sensitive indicator for determining body fatness in children, with triceps skinfold being the most recommended [24]. A review of reviews concluded that BMI should not be applied as the only criterion for reduction of adiposity in children. Other measures such as skinfold thickness are reliable outcomes that can be used to define adiposity status among children [1]. 

Likewise, our study also shows that knowing the process evaluation of an intervention can be of great help in understanding its results. As previously reported, the most serious threat to the effectiveness of an intervention is maintaining the quality of implementation intended by the creators [45]. The degree of implementation of the intervention might affect its results. Thus, fidelity to the established protocol seems to be a key factor in achieving the desired effect. Indeed, the difference in effectiveness according to the fidelity of implementation has previously been demonstrated in other school prevention programs [46]. 

Regarding the differences shown by sex in our results, there is little evidence in the literature that supports or contradicts our findings. Some authors have concluded that there were differences between girls and boys in terms of physiological, psychological, and cultural dimensions that should be taken into consideration before tailoring interventions [1]. Even so, our finding is consistent with the result reported by a previous study, stating that interventions were effective in increasing physical activity at school among boys but not among girls [47]. A possible explanation could be that activities related to the practice of physical activity are more popular among boys than among girls, and that there may be more sports activities on offer to boys. Therefore, interventions that include an effective component in encouraging physical activity could be likely to have a greater effect among boys. 

Our results suggest that the POIBA intervention, carried out in the school environment, including activities in the classroom and in physical education and that also involve the family, might be useful to encourage changes that could prevent obesity among the child population. This is consistent with previous evaluations showing that school-based interventions with a family component and combined especially with actions on physical activity, but also with nutritional elements, could be effective [42,48]. Llargués et al. [49] described a 3.6% decrease in the prevalence of obesity among primary children undergoing an intervention based on promoting healthy eating habits and physical activity at school. A systematic review showed that school-based interventions are generally effective in reducing excess weight in children and may help contain the current increase in childhood obesity [16]. Likewise, there is also a strong consensus on the importance of preventive interventions for childhood obesity at these ages involving the family setting [12,13,48], as families are responsible for feeding their children and also for managing time and activities related to physical activity. 

### Limitations and Strengths

This study has some methodological limitations. One of the main weaknesses of the study could be the lack of comparability between children in the IG and the CG, because participants in these two groups differed in the GDHI index. However, this difference was taken into account when we adjusted for these variables. In addition, among students lost to follow-up, in the IG there were more losses among students attending public and advantaged SES schools compared with the CG. Another limitation was that the sample size was calculated for the whole sample, which did not allow the analysis of the effect stratified by some variables of interest such as sex, SES, or type of school. 

Another possible limitation was that, to better measure adiposity, four skinfolds are recommended, but only one could be measured in our study. Nevertheless, given the impossibility of collecting four skinfolds from a sample of more than 3500 children, we measured the most recommended skinfold. 

Because this was a follow-up study, a common limitation is loss to follow-up. However, more than two-thirds of the children in both groups could be followed up for 3 years and more than half had complete matched data available over the 3-year follow-up, as in previous studies with long-term follow-up [22,26].

Likewise, not all the tools used in the study have a published validation study as in Spain there is a lack of questionnaire validation studies at these ages.

A strength of this study is that it is the first work to obtain data on obesity and to assess a school-based long-term follow-up intervention in a large sample of children in the city of Barcelona. In addition, it is noteworthy that three-year follow-up evaluations with a sample larger than 3500 children are very scarce. 

Another strength is that data were collected through two questionnaires designed “ad hoc” with favorable results to gather information in 8–9 year-old children [31,32]. The good results of the study showed that the children’s age was not a problem for their ability to adequately answer the questionnaires. Our results are in line with some previous studies [50,51]. Likewise, sustainability is a major strength, since the evaluated POIBA intervention has been incorporated into the supply of ASPB school programs, which are freely available to all schools in the city.

## 5. Conclusions

School-based interventions are a good strategy to tackle the global rising childhood obesity. Multilevel and multicomponent school-based interventions, including a family component, could improve children’s healthy habits, especially those regarding food and nutrition and the practice of physical activity. They could also be helpful in preventing the appearance of new cases of childhood obesity, though they may not have an immediate effect on adiposity outcomes.

### 5.1. Implications for Public Health

Interventions like the one evaluated in this study (multilevel and multicomponent) can have an important effect, even if some impact seems to be lost over time. Based on the results obtained, it would be important to go a step further, including the gender and equity perspective in these interventions. It is key to work at a more macro level, developing and implementing structural policies that address social determinants directly linked with obesity and overweight. In this sense, occupational, housing, or social policies might be crucial to minimize the potential inequalities in children. Also, family collaboration is essential and it is a challenge to get the families involved, especially those from disadvantaged socioeconomic backgrounds. The inclusion of sustainable and feasible interventions like the one presented in this study in the school curriculum would be recommended.

### 5.2. Recommendations for Future Studies

Regarding future lines of research, it might be warranted to design evaluations that include big samples and with suitable comparison groups, that would allow an understanding of the real impact of the interventions implemented. In this sense, comprehensive evaluations, including detailed process or implementation evaluation are also needed. Finally, based on our results, there is still an important challenge in the design of interventions that achieve a maintained long-term impact.

## Figures and Tables

**Figure 1 nutrients-13-00453-f001:**
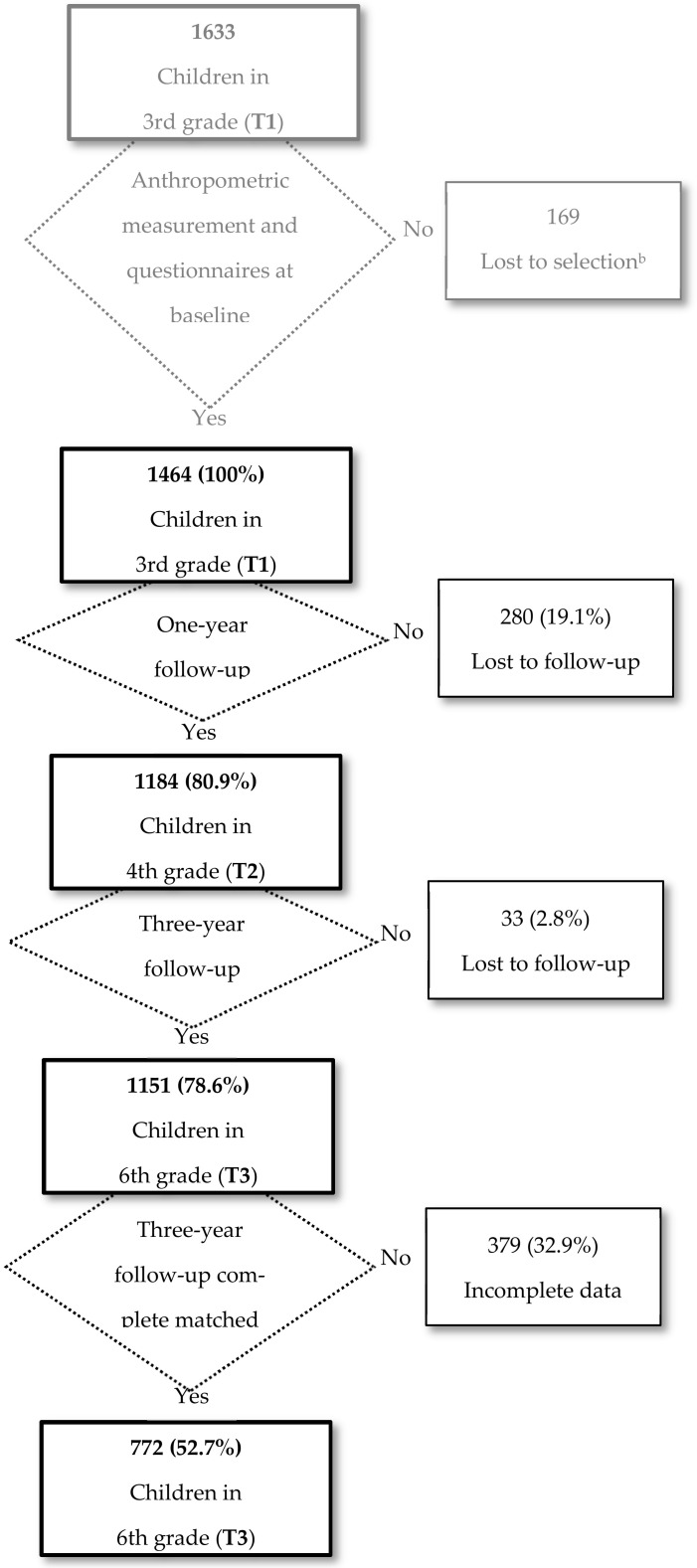
Flowchart of the intervention group participants. POIBA ^a^ Intervention. ^a^ POIBA: Childhood Obesity Prevention in Barcelona (Prevención de la Obesidad Infantil en Barcelona). ^b^ Lost to selection: children for whom no information was available from the questionnaire. T1: baseline. T2: one-year follow-up. T3: three-year follow-up. Grey color: initial total number of eligible participants. Black color: children from whom were obtained anthropometric measurements and questionnaires at baseline.

**Figure 2 nutrients-13-00453-f002:**
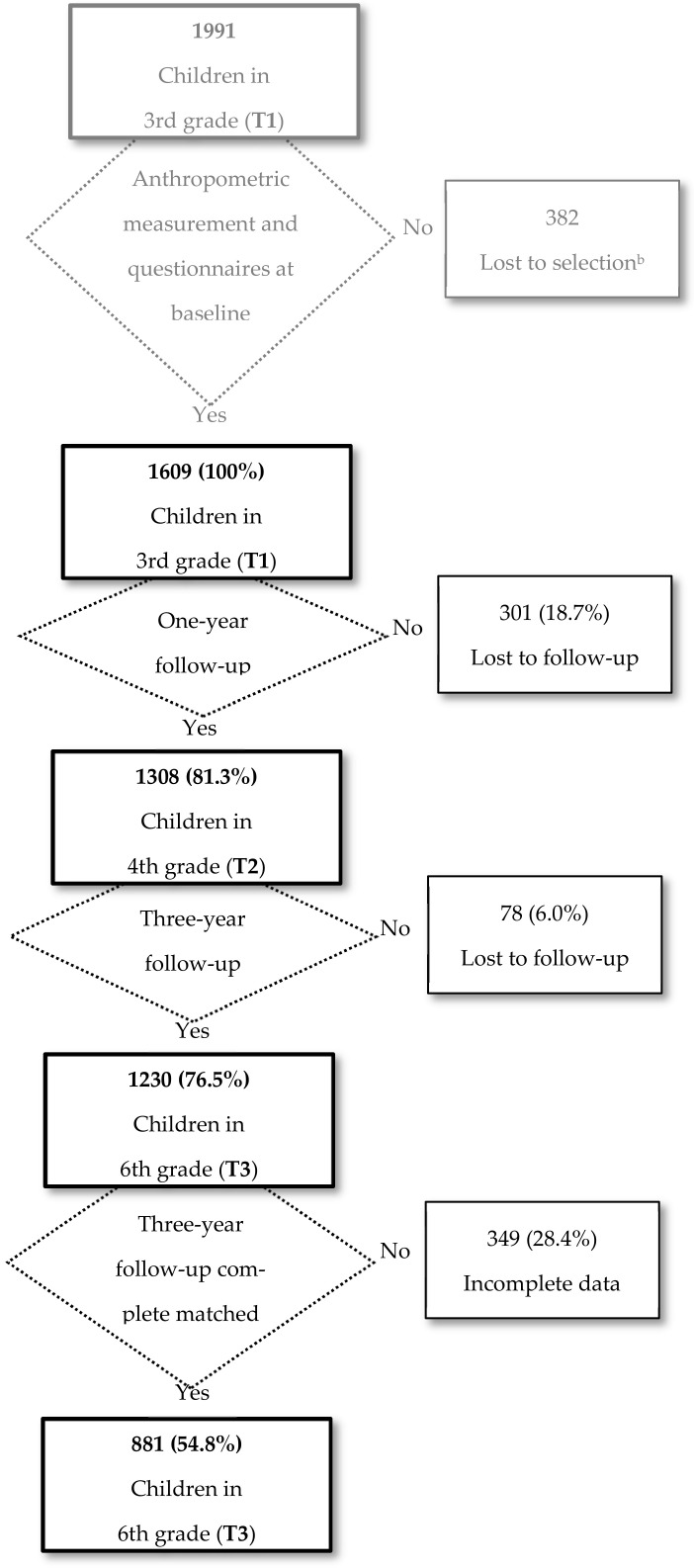
Flowchart of the comparison group participants. POIBA ^a^ Intervention. ^a^ POIBA: Childhood Obesity Prevention in Barcelona (Prevención de la Obesidad Infantil en Barcelona). ^b^ Lost to selection: children for whom no information was available from the questionnaire. T1: Baseline. T2: one-year follow-up. T3: three-year follow-up. Grey color: initial total number of eligible participants. Black color: children from whom were obtained anthropometric measurements and questionnaires at baseline.

**Table 1 nutrients-13-00453-t001:** Sociodemographic characteristics at baseline of children followed at 3 years (*n* = 2681). POIBA ^a^ Intervention. Barcelona, 2014.

	Intervention (*n* = 1151)							Comparison (*n* = 1230)							Total *p* (a) vs. (d)	Complete Cases *p* (b) vs. (e)
	Total (a)		Complete Cases ^e^ (b) (*n* = 772)		Incomplete Cases ^f^ (c) (*n* = 379)		*p*	Total (d)		Complete Cases ^e^ (e) (*n* = 881)		Incomplete Cases ^f^ (f) (*n* = 349)		*p*		
	*n*	%	*n*	%	*n*	%		*n*	%	*n*	%	*n*	%			
**Sex**							0.498							0.714	0.159	0.306
Male	591	51.3	391	50.6	200	52.8		596	48.5	424	48.1	172	49.3			
Female	560	48.7	381	49.4	179	47.2		634	51.5	457	51.9	177	50.7			
**SES** ^b^ **(FAS** ^c^ **Index)**							0.287							0.108	0.158	0.285
High	702	61.0	479	62.0	223	58.8		741	60.2	543	61.6	198	56.7			
Middle	310	26.9	207	26.8	103	27.2		366	29.8	259	29.4	107	30.7			
Low	137	11.9	84	10.9	53	14.0		123	10.0	79	9.0	44	12.6			
Missing	2	0.2	2	0.3	0	0.0		0	0.0	0	0.0	0	0.0			
**Country of origin**							0.023							0.490	0.883	0.556
Natives	795	69.1	550	71.2	245	64.6		853	69.3	616	69.9	237	67.9			
Immigrants	356	30.9	222	28.8	134	35.4		377	30.7	265	30.1	112	32.1			
**Family structure**							0.240							0.835	0.321	0.574
Two-parent home	955	83.0	650	84.1	305	80.5		1,036	84.2	745	84.5	291	83.3			
Single-parent home	180	15.6	111	14.4	69	18.2		186	15.1	130	14.8	56	16.1			
Other	14	1.2	9	1.2	5	1.3		8	0.7	6	0.7	2	0.6			
Missing	2	0.2	2	0.3	0	0.0		0	0.0	0	0.0	0	0.0			
**Type of school**							<0.001							0.742	0.192	0.003
Private/Subsidized	652	56.6	471	61.0	181	47.8		664	54.0	473	53.7	191	54.7			
Public	499	43.4	301	39.0	198	52.2		566	46.0	408	46.3	158	45.3			
**SES (GDHI** ^d^ **Index)**							<0.001							<0.001	<0.001	<0.001
Disadvantaged SES	561	48.7	417	54.0	144	38.0		496	40.3	299	33.9	197	56.4			
Advantaged SES	590	51.3	355	46.0	235	62.0		734	59.7	582	66.1	152	43.6			

**^a^** POIBA: Prevención de la Obesidad Infantil en BArcelona, (Childhood obesity prevention in Barcelona). ^b^ SES, socieconomic status. ^c^ FAS Index, family affluence scale index. ^d^ GDHI Index, gross disposable household income index. ^e^ Complete cases, children with complete data available over the 3-year follow-up. ^f^ Incomplete cases, children with no complete data available over the 3-year follow-up. *p*, *p*-value.

**Table 2 nutrients-13-00453-t002:** Primary and intermediate outcomes at baseline of children followed at 3 years (*n* = 2681). POIBA ^a^ Intervention. Barcelona, 2014.

	Intervention (*n* = 1151)							Comparison (*n* = 1230)							Total *p* (a) vs. (d)	Complete Cases *p* (b) vs. (e)
	Total (a)		Complete Cases^d^ (b) (*n* = 772)		Incomplete Cases^e^ (c) (*n* = 379)		*p*	Total (d)		Complete Cases^d^ (e) (*n* = 881)		Incomplete Cases^e^ (f) (*n* = 349)		*p*		
	*n*	%	*n*	%	*n*	%		*n*	%	*n*	%	*n*	%			
**BMI** ^b^							0.047							0.399	0.167	0.157
Normal weight (<1 SD)	708	61.5	494	64.0	214	56.4		761	61.8	552	62.6	209	59.9			
Overweight (1 SD)	273	23.7	172	22.3	101	26.7		317	25.8	227	25.8	90	25.8			
Obesity (2 SD)	170	14.8	106	13.7	64	16.9		152	12.4	102	11.6	50	14.3			
**Dinner in front of TV**							0.593							0.004	0.634	0.377
No	594	51.6	394	51.0	200	52.8		621	50.5	467	53.0	154	44.1			
Yes	550	47.8	373	48.3	177	46.7		598	48.6	405	46.0	193	55.3			
Missing	7	0.6	5	0.7	2	0.5		11	0.9	9	1.0	2	0.6			
**Eating at school**							0.073							0.001	0.127	0.001
Yes	901	78.3	591	76.5	310	81.8		985	80.1	725	82.3	260	74.5			
No	234	20.3	168	21.8	66	17.4		218	17.7	136	15.4	82	23.5			
Missing	16	1.4	13	1.7	3	0.8		27	2.2	20	2.3	7	2.0			
**Proper breakfast** ^c^							0.113							0.181	0.198	0.936
Yes	447	38.8	312	40.4	135	35.6		508	41.3	353	40.1	155	44.4			
No	697	60.6	455	58.9	242	63.9		711	57.8	519	58.9	192	55.0			
Missing	7	0.6	5	0.7	2	0.5		11	0.9	9	1.0	2	0.6			
**Frequenting fast-food restaurants**							0.911							0.046	0.037	0.286
No	162	14.1	109	14.1	53	14.0		137	11.1	108	12.3	29	8.3			
Yes	977	84.9	653	84.6	324	85.5		1070	87.0	756	85.8	314	90.0			
Missing	12	1.0	10	1.3	2	0.5		23	1.9	17	1.9	6	1.7			
**Eating alone**							0.264							0.997	0.379	0.236
No	1005	87.3	668	86.5	337	88.9		1085	88.2	776	88.1	309	88.5			
Yes	139	12.1	99	12.8	40	10.6		134	10.9	96	10.9	38	10.9			
Missing	7	0.6	5	0.7	2	0.5		11	0.9	9	1.0	2	0.6			
**Out-of-school physical activity**							0.126							<0.001	<0.001	<0.001
Yes	804	69.9	548	71.0	256	67.6		941	76.5	699	79.3	242	69.4			
No	334	29.0	212	27.5	122	32.1		271	22.0	170	19.3	101	28.9			
Missing	13	1.1	12	1.5	1	0.3		18	1.5	12	1.4	6	1.7			
**Leisure time physical activity**							0.247							0.196	0.753	0.809
Yes (≥2 h/week)	740	64.3	485	62.8	255	67.2		782	63.5	551	62.5	231	66.2			
No (<2 h/week)	396	34.4	273	35.4	123	32.5		430	35.0	318	36.1	112	32.1			
Missing	15	1.3	14	1.8	1	0.3		18	1.5	12	1.4	6	1.7			
**Screen time on weekdays** (Monday–Thursday)							0.636							0.003	0.074	0.005
<2 h/day	653	56.7	432	56.0	221	58.3		740	60.2	554	62.8	186	53.3			
≥2 h/day	483	42.0	326	42.2	157	41.4		471	38.3	315	35.8	156	44.7			
Missing	15	1.3	14	1.8	1	0.3		19	1.5	12	1.4	7	2.0			
**Screen time on weekends**							0.112							<0.001	0.023	0.012
(Friday–Sunday)																
<2 h/day	464	40.3	322	41.7	142	37.5		551	44.8	423	48.0	128	36.7			
≥2 h/day	672	58.4	436	56.5	236	62.2		660	53.7	446	50.6	214	61.3			
Missing	15	1.3	14	1.8	1	0.3		19	1.5	12	1.4	7	2.0			

^a^ POIBA, Prevención de la Obesidad Infantil en BArcelona, (Childhood obesity prevention in Barcelona).^b^ BMI, body mass index. ^c^ Proper breakfast, including carbohydrates, dairy and fruits. ^d^ Complete cases, children with complete data available over the 3-year follow-up. ^e^ Incomplete cases, children with no complete data available over the 3-year follow-up. *p*, *p-*value.

**Table 3 nutrients-13-00453-t003:** Changes in food frequency, food behavior, physical activity and daily screen use at three-year follow-up. POIBA ^a^ Intervention. Barcelona, 2014.

		One-Year Follow-Up						Three-Year Follow-Up					
		Sample *n*	Baseline %	Follow-Up %	Unadjusted Change ^e^	Adjusted Differences ^f^ (95% CI)	*p*	Sample *n*	Baseline %	Follow-Up %	Unadjusted Change ^e^	Adjusted Differences ^f^ (95% CI)	*p*
**Food frequency**^b^ (% eating the right number of servings)													
	Water (*4–8 glasses/day*)						0.467						0.767
	Comparison	1308	79.2	82.7	3.5	1		881	79.8	81.8	2.0	1	
	Intervention	1182	76.8	84.0	7.2	1.05 (0.92–1.19)		769	78.8	82.7	3.9	1.02 (0.88–1.20)	
	Fruits and Vegetables (*≥5 pieces/day*)						0.455						0.852
	Comparison	1291	29.1	26.0	−3.1	1		869	28.7	18.0	−10.7	1	
	Intervention	1175	29.3	28.4	−0.9	1.08 (0.87–1.35)		763	30.5	18.6	−11.9	0.97 (0.72–1.31)	
	Meat (*3-4 times/week*)						0.481						0.291
	Comparison	1289	58.4	62.7	4.3	1		868	57.0	54.3	−2.8	1	
	Intervention	1174	57.8	65.3	7.5	1.05 (0.91–1.22)		763	59.5	62.4	2.9	1.10 (0.92–1.32)	
	Occasional sweets (*1 time/month*)						0.278						0.392
	Comparison	1284	54.8	61.1	6.3	1		864	56.1	57.8	1.6	1	
	Intervention	1173	52.3	63.4	11.1	1.09 (0.93–1.26)		762	51.1	57.0	5.9	1.08 (0.90–1.31)	
	Occasionally fried potatoes (*1 time/month*)						0.349						0.512
	Comparison	1286	44.3	54.2	9.9	1		866	45.5	61.4	15.9	1	
	Intervention	1173	40.8	54.0	13.2	1.08 (0.92–1.28)		761	41.1	59.3	18.1	1.07 (0.87–1.30)	
**Food behavior**^b^ (% showing healthy behavior)													
	Proper breakfast (*carbohydrates, dairy and fruits*)						0.330						0.650
	Comparison	1295	41.5	41.3	−0.2	1		872	40.5	36.1	−4.4	1	
	Intervention	1177	39.4	42.8	3.4	1.09 (0.91–1.30)		766	40.7	38.3	−2.5	1.05 (0.84–1.32)	
	Dinner in front of TV (*without screens*)						0.637						0.995
	Comparison	1295	51.7	53.0	1.3	1		872	53.6	50.1	−3.5	1	
	Intervention	1177	51.7	54.9	3.2	1.04 (0.89–1.21)		766	51.3	48.0	−3.3	1.00 (0.82–1.22)	
	Eating at school (*1 healthy meal/day*)						0.625						0.631
	Comparison	1308	82.0	78.1	−3.9	1		881	84.3	73.3	−11.0	1	
	Intervention	1181	80.5	74.3	−6.2	0.97 (0.85–1.10)		769	78.7	65.8	−12.9	0.96 (0.82–1.13)	
	Not eating alone (*better accompanied*)						0.521						0.867
	Comparison	1295	89.3	85.2	−4.1	1		872	89.0	82.6	−6.4	1	
	Intervention	1177	87.8	87.1	−0.7	1.04 (0.92–1.17)		766	87.1	79.8	−7.3	0.99 (0.85–1.15)	
Global Nutrition Score (*% showing > 1 of 9 previous changes*)							0.013						0.188
	Comparison	1308		40.6				881		41.1			
	Intervention	1184		45.5				772		44.3			
Physical activity ^c^ (*% practicing physical activity*)													
	Practicing out-of-school physical activity						0.197						0.364
	Comparison	1283	77.6	79.0	1.4	1		864	80.4	78.2	−2.2	1	
	Intervention	1167	70.6	78.2	7.6	1.09 (0.96–1.24)		740	71.8	75.1	3.3	1.08 (0.92–1.27)	
	Engaging in physical activity in leisure time (*≥2 days/week*)						0.409						0.689
	Comparison	1281	64.3	65.7	1.4	1		864	63.4	72.0	8.6	1	
	Intervention	1165	65.2	70.6	5.4	1.06 (0.92–1.22)		733	63.7	69.9	6.1	0.97 (0.81–1.15)	
Daily use of screens ^d^ (*% using <2 h/day*)													
	On Weekdays						0.290						0.089
	Comparison	1279	61.1	62.9	1.8	1		862	63.9	56.2	−7.8	1	
	Intervention	1164	57.4	63.9	6.5	1.08 (0.93–1.25)		728	56.7	58.4	1.7	1.17 (0.97–1.41)	
	On Weekends						0.379						0.215
	Comparison	1279	45.5	45.8	0.3	1		863	49.0	31.6	−17.4	1	
	Intervention	1164	40.7	44.2	3.5	1.08 (0.91–1.28)		728	42.2	31.5	−10.7	1.16 (0.91–1.46)	
Global Activity Score (*% showing > 1 of 4 previous changes*)							0.009						0.012
	Comparison	1308		15.4				881		11.9			
	Intervention	1184		19.3				772		16.2			
Global Score (*% showing >1 of 13 previous changes*)							0.002						0.025
	Comparison	1308		59.3				881		58.5			
	Intervention	1184		65.3				772		63.9			

^a^ POIBA, Prevención de la Obesidad Infantil en BArcelona, (childhood obesity prevention in Barcelona). ^b^ Spanish nutritional recommendations [36]. ^c^ Physical Activity Guidelines Advisory Committee: Department of Health and Human Services [37]. ^d^ American Academy of Pediatrics [38]. ^e^ Unadjusted change: difference in percentages. ^f^ Adjusted differences: Odds Ratio (OR).

**Table 4 nutrients-13-00453-t004:** Cumulative incidence rate (CIR) of obesity, based on triceps skinfold thickness, by degree of implementation of the intervention and sex (*n* = 1653). POIBA ^a^ Intervention. Barcelona, 2014.

	CIR ^b^ of Obesity							Difference CIR	Ratio of Incidences ^d^	*p*	OR (95% CI) ^e^	Effect Size (*d* Cohen’s)
	Boys		Girls			Total						
	*n* ^c^	%	*n* ^c^	%	*p*	*n* ^c^	%					
Global (1653)	20/652	3.1	38/693	5.5	0.029	58/1345	4.3	-	-	-	-	-
Comparison group (881)	10/332	3.0	18/371	4.9	0.213	28/703	4.0	-	-	-	-	-
Intervention group (772)	10/320	3.1	20/322	6.2	0.064	30/642	4.7	−0.7	−17.3	0.858	0.95 (0.55–1.65)	0.03
Not acceptable	7/135	5.2	10/139	7.2	0.491	17/274	6.2	−2.2	−55.8	0.904	1.04 (0.54–2.00)	0.02
Acceptable intervention group ^f^	3/115	2.6	8/126	6.4	0.165	11/241	4.6	−0.6	−14.6	0.745	0.89 (0.42–1.85)	0.07
Qualified intervention group ^g^	0/70	0.0	2/57	3.5	0.114	2/127	1.6	2.4	60.5	0.107	0.30 (0.07–1.30)	0.67

^a^ POIBA, Childhood Obesity Prevention in Barcelona (Prevención de la Obesidad Infantil en Barcelona). ^b^ CIR, cumulative incidence rate. ^c^ Number of new obesity cases (numerator) among children with normal thickness at baseline (denominator). ^d^ Ratio of incidences (percentage of possible avoided obesity cases). ^e^ OR (95% CI), Odds Ratio and 95% confidence intervals in the multilevel analysis, adjusted by sociodemographic variables: age and sex (individual variables) and type of school and gross disposable household income (GDHI) (contextual variables). ^f^ Acceptable intervention group, the minimum of proposed activities was carried out. ^g^ Qualified intervention group, the intervention was carried out following the established protocol.

## Data Availability

The data presented in this study are available on request from the corresponding author.

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
