# Peer review of "Three-Year Follow-Up of the POIBA Intervention on Childhood Obesity: A Quasi-Experimental Study"

_nutrients, 2021, doi:10.3390/nu13020453_

Round 1

Reviewer 1 Report

The manuscript entitled “Three-year Follow-up of the POIBA Intervention on Childhood Obesity: a Quasi-Experimental Study” presents interesting issue, but some areas must be corrected.

Major:

The main problem with the manuscript is associated with the fact that Authors dis not describe in their manuscript the aspects which are of the highest importance for the readers of “Nutrients” journal, namely the applied programme and its nutritional elements (based on the presented information, reader may assume that food choice recommendations were provided, as Authors assessed Global Nutrition Score). Without the necessary details, the study is of the low value for the readers of the “Nutrients” journal. Moreover, also the discussion should be specific and should include information about the specific dietary modifications.

Abstract:

Authors should briefly describe applied intervention.

Authors should formulate more general conclusions – interesting for all the readers and not formulated only for their specific program.

Introduction:

Authors should present here international perspective – non only Spanish one. If Authors want to publish their manuscript in international journal (not in their national one), they should prepare their Introduction Section to be interesting and valuable for all the readers. Moreover, their manuscript should present important aspects not only for Spanish readers, so should be more universal.

Authors should present proper background for their study – as they studied body mass reduction program for children, they should address similar studies of similar programs and present challenges and problems which are defined by other authors (e.g. https://www.ncbi.nlm.nih.gov/pmc/articles/PMC5634063/; https://www.ncbi.nlm.nih.gov/pmc/articles/PMC7230971/; https://pubmed.ncbi.nlm.nih.gov/22995865/).

Lines 75-77 – should be rather presented in Materials and Methods Section, not in the Introduction Section.

Materials and Methods:

Author should present this section with the necessary details of the program and procedures to be reproducible for other authors.

Authors should provide necessary details of dietary intervention – for the time being there is neither presented how was it planned, nor what it included. Not only general information, but also details are necessary.

Results:

Each table should be stand-alone one (self-explanatory) and Authors should present in the footnotes all necessary details. For the time, for some readers, it may be hard to interpret which statistical test was used in which table, what do abbreviations mean, etc.

Figures – applied colours should be explained in footnotes (grey and black)

Discussion:

Authors should not reproduce in this section the data that were already presented in previous sections.

Authors should in their discussion include 3 areas: (1) compare gathered data with the results by other authors, (2) formulate implications of the results of their study and studies by other authors, (3) formulate the future areas which should be studied.

Conclusions:

Authors should formulate more general conclusions  – being interesting for all the readers and not formulated only for their specific program (not only formulated for POIBA).

Author Contributions:

It seems that contribution of some Authors (SB, GS, SV, OJ) was only minor and they did not participate in preparing manuscript. There is a serious risk of a guest authorship procedure which is forbidden. In such case they should be rather presented in Acknowledgements Section and not be indicated as authors of the study.

Author Response

Dear Mr./Ms.

Yours sincerely,

Francesca Sánchez-Martínez

Reviewer 2 Report

The authors conducted a multicomponent and multilevel school-based intervention on childhood obesity. By comparing intervention group (IG) and the comparison group (CG), the author found that students exposed to the intervention had a significantly better Global Activity score and Global score. The study is well designed. The data is well analyzed and well presented in the manuscript. The results were well discussed. Limitations and strengths were included in the manuscript.

Specifically,

  1. Line 40, please add the reference citation for “overweight or obese in developed countries in 2013.” It would be nicer to cite a more recent reference if there is one. Line 41, please add which year the study of “12.9% for boys and 13.4% for girls” was reported.
  2. Line 76, the “POIBA” is abbreviated for “Childhood Obesity Prevention in Barcelona”, while POIBA is not corresponding to the first letter of “Childhood Obesity Prevention in Barcelona”. Is this a language difference between English and Spanish?
  3. “Data were collected through two validated self-administered computerized questionnaires (line 118)”. The questionnaires were completed by students in the classroom. As all the students aged 8 - 9 years old, I am not sure whether the students at this age can provide precise information in the questionnaires. I think the authors need to consider this issue into their study and consider whether necessary to mention it in “Limitations and strengths” section.

Author Response

Dear Mr./Ms.,

Yours sincerely,

Francesca Sánchez-Martínez

Reviewer 3 Report

The manuscript entitled “Three-year Follow-up of the POIBA Intervention on Childhood Obesity: a Quasi-Experimental Study” presents an interesting issue, however it requires some corrections.

Abstract:

  • Line 24 – “Sweetmeat” - please replace this word
  • Lines 30-31 „The incidence of obesity was lower in schoolchildren when the intervention was implemented in a qualified way, but not statistically significant” – if something is not statistically significant therefore “NO EFFECT EXISTS". Please correct this sentence

Introduction:

  • Line 40 – “developed countries in 2013.” – please provide more up-date reference
  • Line 42 – “aged 5-19 years” individuals over 18 years old is no more child (or adolescent) but adults.
  • Line 52 – “not being breastfed. [5].” It should be “not being breastfed [5].”
  • Line 55 – “The stabilization of BMI values” – I am not a native speaker of English but the English language should be improved .
  • Lines 67-68 – “In the “Healthy Schools” project, these effects were combined with greater consumption of fruits and vegetables [19].” – some information could be presented in a more detailed way. Please specify this project and the results of this project.

Materials and Methods:

  • Line 85 – “schoolchildren” – it should be “children”
  • Line 112 - 2.4. Anthropometric Measurements” – please provide the criteria (cutoff) for overweight and obesity in children (percentile grids? – please provide specific reference)
  • Line 118-120 – “Data were collected through two validated self-administered computerized questionnaires, “POIBA-How do we eat?” for food frequency and food behavior and “POIBA-How do we move?” for physical activity and screen use [25, 26]. “ – More information is needed about the validity and reliability of each measure. Additionally, any limitations in reliability and validity need to be addressed in the discussion. This is crucial, especially for “POIBA - How do we move?” questionnaire (no publications associated with the validation are provided).
  • Lines 148-155 –reference is needed
  • Lines 160-161 – “the Family Affluence Scale 160 (FAS) [33].” - What is the original language of the questionnaire? Original language English? Was the questionnaire translated? Who did so? Any validation of the translated questionnaire? More information is needed.
  • Line 162 – “Disposable Household Income (GDHI) (disadvantaged SES for <85/advantaged SES for ≥85)[34]. “ what is the unit?
  • The information about the intervention should be presented in detail.
  • “Statistical Analysis“ section - The information about α (p-Value) is needed.
  • The number of children at baseline in table 1 (title) and flowchart is differ. Authors should be more consistent. In table 1 authors presented data from t3? But the title is “at baseline”. Please unity it.

Results and Discussion:

  • Line 445 – “benefits [16, 17] Nevertheless,” it should be “benefits [16,17]. Nevertheless,”
  • Line 463 – “literature [38, 39] BMI is” it should be “literature [38,39]. BMI is”
  • Line 522 – “children [25, 26] Likewise” it should be “children [25,26]. Likewise”
  • Line 522 – “children [25, 26] Likewise” it should be “children [25,26]. Likewise”
  • Lines 520-522 – “Another strength is that data were collected through two questionnaires designed “ad hoc” and validated with favourable results, as there were no previously validated questionnaires to gather the information in 8-9 year-old children [25, 26]” – the reference no. 26 is not a publication – as it was mentioned above – More information is needed about the validity and reliability of each measure. Additionally, any limitations in reliability and validity need to be addressed in the discussion. This is crucial, especially for “POIBA - How do we move?” questionnaire (no publications associated with the validation are provided).

Conclusion:

  • Line 531 – “The intervention also showed a positive trend in decreasing” – if it is not statistically significant therefore “NO EFFECT EXISTS". Please correct this sentence. I think the sentence from discussion “Unlike the results at the 1-year follow-up, when the effect of the intervention was significant [23], the effect diminished at the 3-year follow-up, losing significance but maintaining the positive trend.” will be more suitable as a conclusion.
  • Moreover, authors should focus (in discussion section) on the barrier of maintaining the effect of the intervention. This could be very interesting for readers.

Author Response

(The authors gave the same response as above.)

Round 2

Reviewer 1 Report

The manuscript entitled “Three-year Follow-up of the POIBA Intervention on Childhood Obesity: a Quasi-Experimental Study” presents interesting issue, but some areas must be corrected.

Major:

The main problem with the manuscript is associated with the fact that Authors still did not describe in their manuscript the aspects which are of the highest importance for the readers of “Nutrients” journal, namely the nutritional elements of the program. The program itself is now properly presented, but still we do not see the deepen information about nutritional elements. Without the necessary details, the study is of the low value for the readers of the “Nutrients” journal.

Abstract:

Authors should briefly describe applied intervention (nutritional elements of the program)

Materials and Methods:

Authors should provide necessary details of dietary element of the program – for the time being there is neither presented how was it planned, nor what it included. Not only general information, but also details are necessary.

Author Response

Dear Sir/Madam,

Kind regards,

Francesca Sánchez-Martínez

Reviewer 3 Report

I appreciate the great efforts that the authors have made in response to my questions and concerns. However, there are some issues that should be corrected:

For the validation of food frequency questionnaires, there are a specific recommendations (Cade et al, https://www.ncbi.nlm.nih.gov/pubmed/19079912), that should had been applied.

According to the recommendations of Cade, the specific methods should be applied in the validation studies and validation studies are necessary to conduct assessment. The analysis of correlation is not the recommended method (so Authors should not conclude on the basis of it). At the same time, the kappa statistic and Bland-Altman method are the recommended methods. They should calculate the Bland-Altman index (in %) and conclude on the basis of the commonly indicated criteria (e.g. presented by Myles & Cui, https://www.ncbi.nlm.nih.gov/pubmed/17702826).

Due to the fact, that this is not the main aim of the study, I think these new analysis could be omitted. However authors must:

  • Removed sentences in lines 520-524 – Due to the issues associated with the validation of the questionnaire – theses sentences should be removed.
  • Add to the limitation section the information about the lack of validation of some tools (e.g. “there are no published studies on the validation of the questionnaire in Spanish language”

Author Response

(The authors gave the same response as above.)
